# Study on High-Temperature Interaction Mechanism of Nd–Fe–As System

**DOI:** 10.3390/ma12193060

**Published:** 2019-09-20

**Authors:** Chenghui Fu, Run Huang, Wenhao Xie, Jinxiao Luo, Yulian Li, Jinzhu Zhang

**Affiliations:** 1School of Materials and Metallurgy, Guizhou University, Guiyang 550025, China; gzu_fch@163.com (C.F.); rhuang@gzu.edu.cn (R.H.);; 2Guizhou Province Key Laboratory of Metallurgical Engineering and Energy Process Saving, Guiyang 550025, China

**Keywords:** Nd, Fe, As, compounds

## Abstract

In this study, Nd and As are mainly sealed into industrial pure Fe cylinders. The effect of different temperatures on the high-temperature interaction of an Nd–Fe–As ternary system is studied via X-ray diffraction, optical microscopy, and scanning electron microscopy after heat insulation for 30 h at 1173, 1223, and 1273 K. The results show that the common products under high-temperature interaction are NdAs, Fe_17_Nd_2_, and Fe. Fe_12_As_5_ is present at 1173 K, whereas Fe_2_As is produced at 1223 and 1273 K. The diffusion ability of Nd is weaker than that of As. Nd mainly diffuses through the Fe atomic vacancy mechanism. As mainly binds to Fe to form Fe and As compounds.

## 1. Introduction

With the progress of science and technology and the continuous need for social development, the world’s steel output has reached 2 billion tons per year. Given the sharp increase in steel production, the scrap cycle is becoming increasingly short. In the recycling process of scrap, Sn, Pb, As, and other low-melting-point impurities are inevitably brought into the process. Low-melting-point impurities such as Sn, Pb, As and the like, cannot be effectively removed in conventional steel making [1,2]. Given the strong chemical activity of rare earth elements, enrichment of the low-melting-point impurity As, as well as other elements in steel, is reduced or eliminated [3,4]. As is a silver-white metal with an atomic number of 60, a melting point of 1297 K, and a density of 7.004 g/cm^3^; it exhibits paramagnetism and has a relatively high chemical activity, such that the surface tends to oxidize upon prolonged exposure to air to a dark gray color. Arsenic is solid at room temperature, and appears grayish white with an evident metal luster. Its sublimation occurs at 888 K, with a garlic-like smell, and it is colorless and highly toxic. The melting point of As is 1087 K at 35.8 atmospheric pressure. Therefore, in general, As changes directly from a solid state to a gaseous state without a melting state.

Arsenic has a degree of segregation of up to 42 in carbon steel [5], which can be present in the form of Fe_2_As, Fe_3_As_2_, FeAs, and as a solid solution. It tends to converge at the interface, deteriorating the thermoplastic of a casting billet. With increasing As content, steel exhibits decreased plasticity and impact toughness. Given its hot brittleness, the material is scrapped, or the low-temperature tempering brittleness causes the material to fail in the course of service, often leading to the disastrous loss of property and life [6,7,8]. The binary phase diagram of Nd–Fe has been previously summarized [9], and the stable compounds can be formed between the rare earth elements and Fe, i.e., Fe_17_Nd_2_ and Fe_2_Nd. According to the Fe–As binary phase diagram, the likely compounds are: Fe_2_As, Fe_3_As_2_, FeAs, and FeAs_2_. The solubility of As in α-Fe has a maximum value of 10% approximately at 1113 K [10]. With decreasing temperature, the solubility of As in Fe decreases gradually, and the solubility of As in Fe is less than 5% at room temperature [11,12,13]. Different atomic ratios of cerium (lanthanum), Fe, and As can form the ternary compounds RE_12_Fe_57.5_As_41_ (RE=La, Ce) and FeAs at 1173 K [14,15]. The ternary compound La_10_Fe_50_As_40_ is formed by the La–As–Fe system at 1223 K [16]. The high-temperature interaction of Nd–Fe–As has been studied by Wenhao [17], and the existence of the NdAsFe ternary compound may be present. In addition, NdAs and FeAs_2_ are likely to be prerequisites for the formation of the ternary compound NdFeAs.

On the basis of this finding, a certain quality of Nd and As seals are heated to 1173, 1223, and 1273 K for 30 h in a special cylinder block processed by industrial pure Fe. The interaction among Nd, Fe, and As at low-melting point is studied via optical microscopy, scanning electron microscopy, and X-ray diffraction (XRD).

## 2. Materials and Methods

Figure 1 shows a barrel-shaped cylinder that is composed of industrial pure Fe, with the following main chemical composition (mass fraction): 0.02% Mn, 0.005% Al, 0.002% C, 0.006% P, and 0.004% S. The oxide film on the Nd surface was removed, and the granulated rare earth Nd and As were weighed on the basis of a certain mass and filled into the cylinder. The screw plug was then tightened, and the cylinder block was sealed by arc welding. The sample was then placed into a closed SRJK-2-9 tube-type vacuum resistance furnace. Ar was used as the protective gas for heating. The specific temperature is shown in Table 1.

Table 2 shows that the experimental heating process was established on the basis of the vapor pressure of As. After heating to 1173, 1223, and 1273 K, the temperature was maintained for 30 h. Finally, the furnace naturally cooled to room temperature after the power was turned off and the protective gas was stopped. The barrel-shaped cylinder samples were divided into two parts by using a hand hacksaw at a 16-mm distance from the bottom of the samples. One part was processed into a metallographic sample. The infiltrates of Nd, As, and Fe after the high-temperature treatment were separated from the center of the other part. After crushing and grinding the samples, the phase composition was analyzed using a PHILIPS X-PERT PRO diffraction instrument. The XRD experimental parameters were as follows: Cu target, λ = 0.154056 nm, 40 kV voltage, and 2°/min scanning speed.

## 3. Results and Discussion

### 3.1. Metallographic Analysis

Figure 2 shows the metallographic pictures of different temperature samples (1, 2, and 3, from left to right) under the optical microscope. Three types of contrast (light gray, gray, and black) were observed at different temperature conditions. In sample 1, the light gray phase has a higher segregation than the rest; however, it is not evenly distributed in the grayish matrix. The light phase segregation is lower in image two, as the matrix is becoming finer. In image three, the dark phase becomes more uniformly distributed in a fine matrix in the gray phase and the evenly spread light gray phase. In samples 1, 2, and 3, the distribution of different contrast degrees is similar, and the three kinds of contrast distribution of sample 3 are the most uniform. At this time, the distribution area of the dark phase is enlarged, whereas the gray phase is gradually decreased.

### 3.2. Analysis of Elemental Distribution in Sample

Figure 3 shows the morphology and element distribution of the sample at different temperatures. The dark gray contrast area is represented by A1, A2, and A3; the light gray contrast area is represented by B1, B2, and B3; and the gray contrast area is represented by C1, C2, and C3. Arsenic is distributed in the whole sample, and a small part of Nd and Fe coexist. According to the analysis of the Energy Dispersive Spectroscopy (EDS) results of backscatter electron images and samples, the white area is the NdAs phase, the light gray area is Fe_17_Nd_2_, and the gray area is Fe_12_As_5_. At 1223 K, the white area is the NdAs phase, the light gray area is Fe_17_Nd_2_, and the gray area is Fe_2_As. The contrast at 1273 K is consistent with that at 1223 K.

As shown in Figure 3, the Nd has a portion that enters the cylinder, indicating that the diffusion capacity of the metal is weaker than that of As. With increased temperature, the reaction of the Nd atom is promoted. When the temperature rises to different temperatures, the melting point of Nd is 1297 K, due to the sublimation of As at 888 K. The As in the sample then exists in the form of gas, whereas Nd exists in the liquid or solid phase.

### 3.3. Phase Analysis of Samples

Figure 4 shows the XRD pattern of the sample at different temperatures. The diagram illustrates that the high-temperature products are NdAs, Fe_17_Nd_2_, Fe_12_As_5_, Fe, and Fe_2_As. With increased temperature, the products of 1223 and 1273 K remain the same. At 1173 K, no Fe_2_As occurs, but Fe_12_As_5_ is present, indicating that when the atomic ratio of Nd to As is 1.0:2.0 and the temperature is higher than 1173 K, the Fe_2_As phase disappears, but the Fe_12_As_5_ phase is formed. According to the iron–carbon phase diagram, γ-Fe can be converted to α-Fe when the temperature is lower than 1185 K. With the increase in temperature, the characteristic peak intensity of 1273 K of the XRD pattern is lowest at 46.5°, whereas the intensity of the characteristic peak increases gradually when the characteristic peak intensity of the XRD pattern is at 44.5°. An undetermined diffraction peak is also observed in the XRD pattern. According to the results of EDS analysis and the literature report of Stoyko, the unknown diffraction peak may be the ternary compound RE_12_Fe_57.5_As_41_ [14].

The crystallite size of different temperatures was estimated by the Debye–Scherer equation [18]. Table 3 demonstrates the results.
(1)$$D=\frac{\kappa \lambda }{\beta \cos\theta },$$

In this equation, *β* is full width at half maximum (FWHM), *κ* is Scherer’s constant (usually equal to 0.9), D is the crystallite size, *λ* is the wavelength of the X-ray, and *θ* is the angle of diffraction. With the increase of temperature, the crystallite size of the sample becomes smaller. Accordingly, a high fraction of the boundary regions exists between the phases (it cannot be a single phase for boundaries to coexist). These boundaries have highly disordered structures which increase the reactivity of the phases. With the increase of temperature, the crystallite size of the sample becomes smaller. Since the radius of the Nd atom is larger than that of the Fe and As atoms, the vacancy number of Fe atoms is formed through thermal vibration. Therefore, the vacancy number increases with an increase in temperature. On the contrary, Nd tends to accumulate at the grain boundaries easily, preventing the grains from growing, and so the grains are smaller.

### 3.4. Analysis of Elemental Distribution in Sample

Figure 5 shows the line analysis of the transition region of the sample at different temperatures. The three elements of Nd, Fe, and As have a continuous band-like gradient distribution. The content of Nd is the highest in the white contrast area, the content of Fe is the highest in the black contrast area, and As exists in all phases. With the increase of temperature, the distribution of the Fe elements in the matrix becomes gradually uniform, the diffusion of As elements in the matrix does not change obviously, while the Nd accumulates gradually at the boundary. The diffusion of As mainly occurs through the formation of Fe and As compounds, while the diffusion of Nd occurs through the vacancy mechanism and the formation of NdAs compounds. As the Nd will have a higher chemical potential as the temperature is raised and since As only has 10% solubility at such high temperatures, vacancies will be generated during sublimation. First, As adsorbs and then reacts with Nd to form NdAs-type compounds, and then the remaining Nd forms compounds with the Fe to form Fe_17_Nd_2_ upon the generation of sublimation-derived vacancies. The vacancy number of the Fe atoms is formed through thermal vibration. Furthermore, the vacancy number increases with increased temperature. These vacancy sites provide conditions for the diffusion of Nd atoms [19]. With the increase in temperature, as the grain size becomes smaller, the grain boundary area increases, and there are more vacancies at the grain boundary. The combination of the vacancy and the Nd atom leads to the formation of the solute–vacancy complex, such that the thermally enhanced diffusion rate of the boundaries causes the Nd atoms to segregate in the grain boundary region.

## 4. Conclusions

(1) Under high-temperature interaction, the products of 1173 K are NdAs, Fe_17_Nd_2_, Fe, and Fe_12_As_5_. The products of 1223 K and 1273 K are NdAs, Fe_17_Nd_2_, Fe, and Fe_2_As.

(2) With the increase of temperature, the crystallite size of the sample becomes smaller.

## Figures and Tables

**Figure 1 materials-12-03060-f001:**
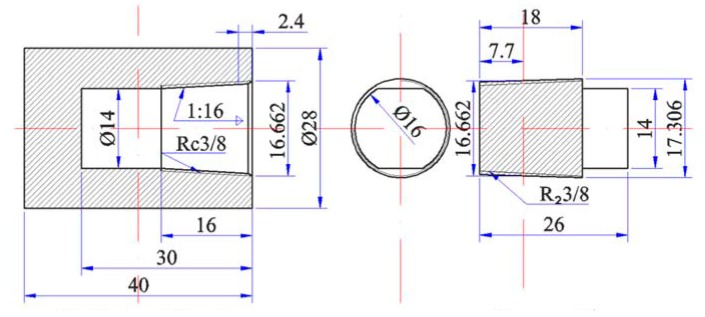
Industrial pure iron cylinder (mm).

**Figure 2 materials-12-03060-f002:**
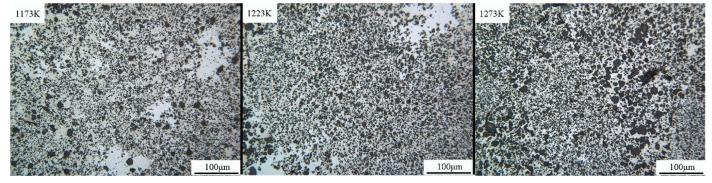
The metallographic pictures of samples with different temperature (Nd:As = 1:2, t = 30 h).

**Figure 3 materials-12-03060-f003:**
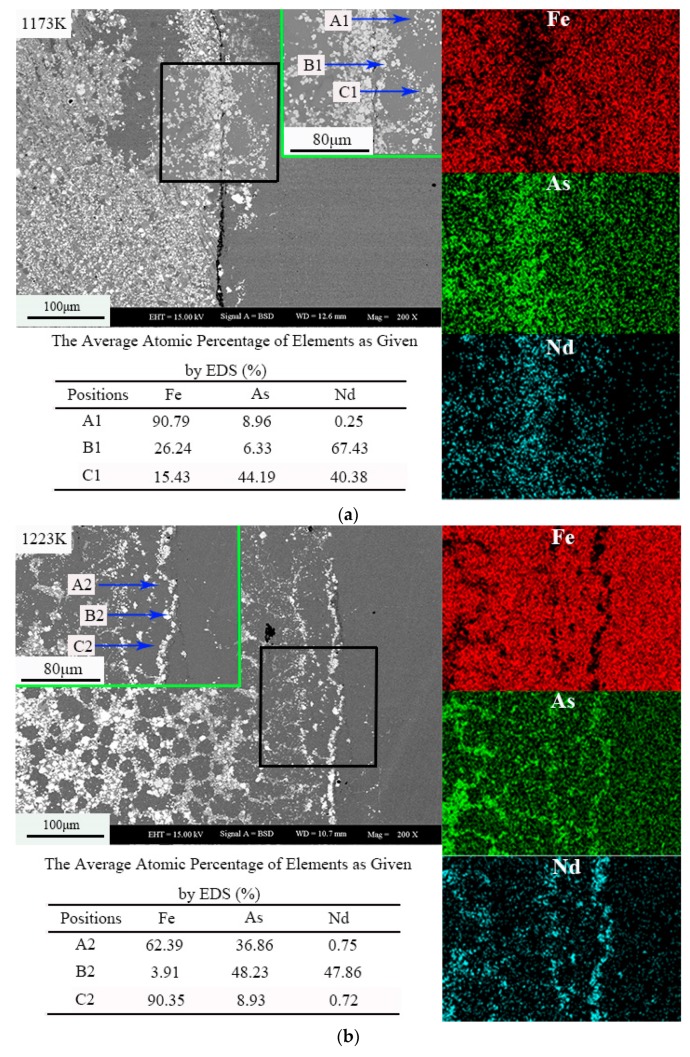
The morphology and element distribution of the sample (Nd:As = 1:2, t = 30 h), (**a**) 1173 K, (**b**) 1223 K, (**c**) 1273 K.

**Figure 4 materials-12-03060-f004:**
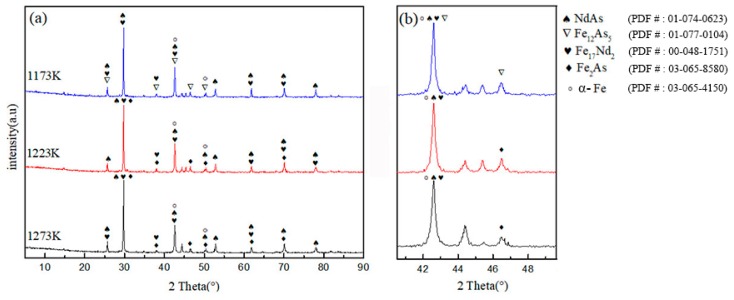
(**a**) X-ray diffraction patterns of samples, (**b**) a partial magnification of the diffraction angle 2θ at 40° to 50°.

**Figure 5 materials-12-03060-f005:**
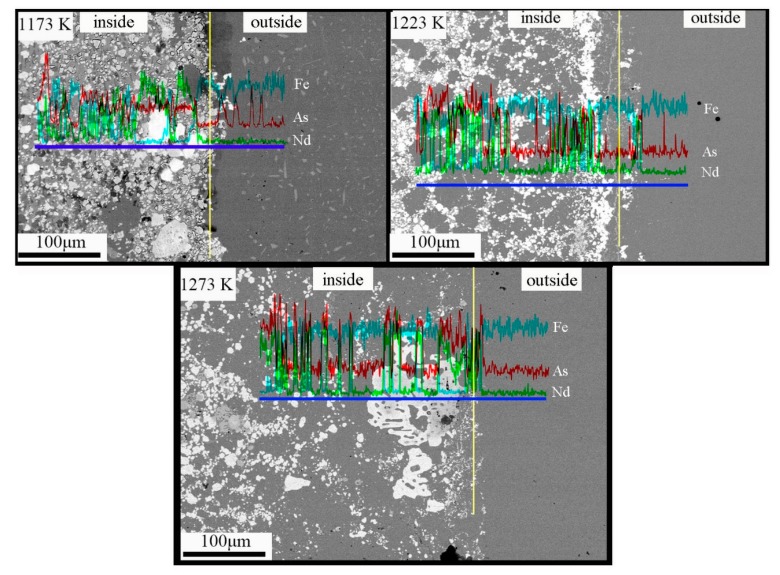
The line analysis of the transition area of sample (Nd:As = 1:2, t = 30 h).

**Table 1 materials-12-03060-t001:** Sample experimental parameters.

Sample	Atomic Ratio	Nd (g)	As (g)	T (K)	T (h)
1#	1.0:2.0	4.9014	5.0985	1173	30
2#	1.0:2.0	4.9014	5.0985	1223	30
3#	1.0:2.0	4.9014	5.0985	1273	30

**Table 2 materials-12-03060-t002:** Sample heating parameter.

Experimental Heating Process
$1\#\;\mathrm{Room}\;\mathrm{temperature}\overset{2h}{\to }773K\overset{10K/10\min}{\to }923K\overset{10K/20\min}{\to }983K\overset{10K/30\min}{\to }$ $1023K\overset{10K/1h}{\to }1073K\overset{10K/2h}{\to }1123K\overset{10K/5h}{\to }1173K(30h)$ $2\#\;\mathrm{Room}\;\mathrm{temperature}\overset{2h\;}{\to }773K\overset{10K/10\min}{\to }923K\overset{10K/20\min}{\to }983K\overset{10K/30\min}{\to }$ $1023K\overset{10K/1h}{\to }1073K\overset{10K/2h}{\to }1123K\overset{10K/5h}{\to }1173K\overset{10K/6h}{\to }1223K(30h)$ $3\#\;\mathrm{Room}\;\mathrm{temperature}\overset{2h\;}{\to }773K\overset{10K/10\min}{\to }923K\overset{10K/20\min}{\to }983K\overset{10K/30\min}{\to }$ $1023K\overset{10K/1h}{\to }1073K\overset{10K/2h}{\to }1123K\overset{10K/5h}{\to }1173K\overset{10K/6h}{\to }1223K$ $\overset{10K/7h}{\to }1273K(30h)$

**Table 3 materials-12-03060-t003:** Crystallite size.

T	1173 K	1223 K	1273 K
FWHM (°)	0.171	0.186	0.195
Crystallite Size (nm)	48	44	42

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
