# Peer review of "Study on High-Temperature Interaction Mechanism of Nd–Fe–As System"

_materials, 2019, doi:10.3390/ma12193060_

Round 1
Reviewer 1 Report
The paper has an important content from a point of view of material science. So it perfectly fits to the journal profile. However the idea of the paper is not entirely new (even if very interesting), because behavior of impurities in some materials, especially as important as steel is broadly done. The novelty of the paper is included in the type of impurities investigated. The paper has some small drawbacks, which I think would be good to eliminate. First, the procedure of heating is shown in details, but the cooling process not. Probably Authors are sure that the cooling process did not influence the product. Even if it is like that an information should be given what was the pace of cooling and why Authors are sure the products are not changed. Presenting XRD data you write about boundary disordered structure. One or two sentence would be necessary to explain, why you think you observe order-disorder within the boundary region. One of the most important conclusions is that the mechanism of elements diffusion is a diffusion through vacancies. What is the proof ?? The Authors cited a previous work and this is OK, however you can’t give a previously recognized mechanism as a result / conclusion for the present paper. If there would be done a new research e.g. FIM, STM measurements giving arguments in favor, you might use it for your conclusion. But maybe you have dynamic recovery and recrystallization (DRV) involving dislocation annihilations and rearrangement into low-energy subgrain boundaries (SGB). Maybe your need TEM to exclude such a possibility ?? Your conclusion is not justified / obtained due to your present research. The conclusion is assumed that the mechanism is the same as previously recognized. To do measurement and to get measurements results and to assume results are two different matters. So please append your research or write that you assumed previously recognized solution. But if you assumed, this can’t be your conclusion. It is important your conclusions are drawn from your present research, not from previous ones. If you assumed the mechanism it also is important to make sure, additional research was not necessary.
Author Response
Dear Professor ,
We are grateful to the editor and the two referees for their constructive comments and suggestions on the revision of the manuscript again. We have made all the necessary changes as suggested by the referees. All the revisions in the manuscript and the revised information have been highlighted in red color.
Now, we are submitting the revised version of our manuscript. A detailed list of our responses to the referees’ reports is also given following this page.
Best regards,
Jinzhu Zhang

Reviewer 2 Report
Dear authors it was a decent presentation of results but elaboration may be required in several areas of experimental part well justified with discussion.
Please amend and correct the following:
Line 24: cycle of scrap cycle? Scrap cycle may be…
Line: 29 - 33: silver-white metal with an atomic number of 60, melting point of 1297 K, and density of 7.004 g/cm3. It has paramagnetism, a relatively active chemical property, and a surface that can be oxidized to dark gray upon prolonged exposure to air. As is solid at room temperature and appears grayish white with an evident metal luster. Its sublimation occurs at 888 K, and its steam has a garlic smell and is colorless and highly toxic. The melting point of As is 1087 K at 35.8 atmospheric pressure.
Can be changed to following:
"silver-white metal with an atomic number of 60, melting point of 1297 K, and density of 7.004 g/cm3, exhibits paramagnetism and has a relatively high chemical activity such that the surface tends to oxidize upon prolonged exposure to air to dark gray color. Arsenic is solid at room temperature and appears greyish white with an evident metal luster. Its sublimation occurs at 888 K, with garlic-like smell, it is colorless and highly toxic. The melting point of As is 1087 K at 35.8 atmospheric pressure."
Line 35: Please explain for the audience and elaborate more on: As has a degree of segregation of up to 42.
What is degree of segregation in alloys? In what system we are talking about? Please when you start the sentence with Arsenic, use full form instead of As.
Line 41 - 44: compounds can be formed between the rare earth elements and Fe, which are the two compounds of Fe17Nd2 and Fe2Nd, respectively. According to the Fe-As binary phase diagram, the Fe2As, Fe3As2, FeAs, and FeAs2 binary compounds can be generated after Fe and As are combined. When the solubility of As in α-Fe is 1113 K, the maximum value is approximately 10%.
"compounds can be formed between the rare earth elements and Fe, i.e. Nd2Fe17 and NdFe2. According to the Fe-As binary phase diagram, the likely compounds are: Fe2As, Fe3As2, FeAs, and FeAs2. The solubility of As in α-Fe has a maximum value of 10% approximately at 1113 K."
Line 59: The rare earth Nd and As were weighed on the basis of a certain mass and filled into the cylinder.
How and in what form Nd and As were added to the cylinder, as the rare-earths have high affinity for oxygen? What were the preparation conditions? Please elaborate.
Line 79: Figure 2 shows the gold phase diagram of different temperature samples under the optical.
What do you mean by gold phase diagram?
Line 81 - 83: In sample 1, the light gray area is evenly distributed in the whole sample, gray is the base of the sample, and the black area is relatively small in the whole sample.
"In sample 1, light grey phase has higher segregation than the rest but it is not evenly distributed in the greyish matrix. Light phase segregation is lower in image 2 as the matrix is becoming finer. In image three, dark phase becomes more uniformly distributed in a fine matrix of grey phase and evenly spread light grey phase."
Is it possible to explain which phases may appear of what morphology and color contrast in optical microscopy?
Line 85 - 86: At this time, the distribution area of black area is enlarged, whereas the gray area is 85 gradually decreased.
"Slightly better selection of words will improve the sentence quality."
Line 89: 3.2. Phase analysis of samples
"I would actually suggest to incorporate FEG-SEM results as Section 3.2 to compare with the optical microscopy and then correlate the results with phase analysis to match EDS composition."
For the moment, optical microscopy images seem to have no representative comparison with FEG-SEM as the images are only relative to small area of analysis. It would be better if comparison is drawn between optical microscopy and SEM.
Line 111: 3.3. Analysis of elemental distribution in sample
I must say, the SEM image representation is not very clear. The copied images are of low resolution and fine details of the microstructure cannot be seen.
The propensity of light phase increase with the temperature which is directly related to Nd vacancy compounds formation at elevated temperatures. But these phases are not quantified yet with the EDS. The phases are large enough to be precisely determined with the EDS at 15 - 20 kV. In comparison to Nd2Fe17 phase the NdAs phase has higher volume fraction in the representative micrographs. This white phase tends to segregate as the temperature is increased and darker Fe matrix becomes more uniform.
What is the reason for such morphological changes?
The average atomic percentage of elements correspond to which region? The green inset is unclear and EDS quantification is absent to support the results for phases identified.
Line 122 - 123: Given that gas–solid diffusion is stronger than 122 liquid–solid diffusion, Nd is more difficult to spread into a cylinder block compared with As.
Why is Nd becoming part of the matrix as compared to As-Fe based compounds if he thermal diffusivity of As is higher? This is not true.
"Nd will have higher chemical potential as the temperature is raised and since As only has 10% solubility at such high temperatures, vacancies will be generated during sublimation. First Nd will react with As to form NdAs-type compounds and then the remaining Nd will form compounds with Fe upon the generation of sublimation derived vacancies. The grain boundaries play important role in the formation of Nd2Fe17 phase whereas gas phase adsorption and condensation will formulate NdAs phase. Please bear in mind, it is As which adsorbs and then reacts with Nd, so although larger diameter Nd atoms can only progress through lattice vacancies its diffusion may be limited to sublimation rate but not the chemical potential (which is still higher than Fe and As)."
I would suggest enthalpies of formation to be known for such sublimation-condensation reactions.
Besides the impact of grain boundaries and their enrichment with Nd has not been considered. In case of sample 1, the grain boundary surface area is higher than sample 3, where precipitates begin to form solely because of As sublimation. If there would have been a competition in reactions (in the absence of sublimation), Nd would spread faster via grain boundary vacancy flipping due to thermal vibrations rather than consolidation to Fe matrix as Nd2Fe17 phase.
Line 126: 3.4. Analysis of elemental distribution in sample
Now here the things get interesting. In image 1 or sample 1, we can clearly see in the outside zone new kind of platelet shaped precipitates originating within the Fe matrix with slightly brighter contrast. What is diffusing outside and forming up the compounds with Fe here at lower temperatures? These precipitates tend to disappear in sample 2 and 3 outside zone, so what inhibits phase transition at higher temperatures and importantly an alternate reaction may be trapping such a specie.
Moreover, if we look at Figure 4, the grey phase is distributed very finely in the inside zone of sample 1, i.e. lower temperature. This phase tends to grow and encircle the darker greyish phase as in case of sample 2 and 3. Which phase is this? There are two different contrasts of fine grey phases here as well. Do you know which are they? If the brighter contrast is more evenly spread, it may be considered that Nd is diffusing e.g. sample 2, sparsely in sample 1 as the bright aggregates are larger.
If you look at the interface, there is a dark grey phase at the top part of sample 1. It should contain more Fe. This phase also disappears at the interface of sample 2 and 3. Do we expect some residual Fe left over at his temperature range in sample 1, such that vacancies are drawn as As sublimes (partially) and Nd atoms oscillate to these sites forming brighter contrast phases. The areal fraction of grey and dark grey phase is significantly less in image 2, what might be happening here as bright phase is spread across?
Darker or blackish phase also appear in sample 1 of Figure 5. Seldomly this phase appears in sample 3, but higher contrast means it is composed more of Fe.
It is advised that EDS phase quantification is performed on all the phases in 3 samples and presented to signify the discussion and compliment the line scan. At the moment, this is not very clear.
Line 89: 3.2. Phase analysis of samples
Let’s quickly go through the XRD analysis. I am curious why the Fe matrix of the barrel cylinder is not detected in the XRD? By order of preference and increasing the As content, following phases shall develop in Fe-As binary alloys: Fe2As, Fe3As2, Fe12As5, FeAs and FeAs2. So the analysis indicated Fe-rich Fe2As and intermediate phase presence. Mention the JCPDS or XRD reference cards for the phases identified.
NdAs did form in sample 1? Your conclusions say otherwise.
I am not sure, if NdAs, Nd2Fe17 and Fe12As5 can have same peak positions. Elaborate that NdAs peak positions are more visible in sample 2 and 3 over Fe2As but give reference peak positions indication or maybe you can show them as well. There can also be presence of NdFe4 compounds which may match your peaks. Any indication of ternary phases?
Determine with EDS analysis the different phases that might be present in the system at different temperatures.
Line 106 - 109: With the increase of temperature, the crystallite size of the sample becomes smaller and smaller. Accordingly, high fraction of the boundary regions existed between Phase. These boundaries had a highly disordered structure which increased reactivity of the Phase.
Which crystallite are we talking here?
“the crystallite size of the sample becomes smaller”
“high fraction of the boundary regions existed between phase” = which phase and why reactivity of this phase increases?
Line 110: Table 3 Crystallite Size
Why the crystallite size decreases with the temperature?
Line 141: The diffusion ability of Nd is weaker than that of Fe and As
Fe will not contribute to diffusion quite likely as it is the matrix.
Mention different phases identified at temperature ranges as well in conclusions.

Author Response

(The authors gave the same response as above.)

Round 2
Reviewer 2 Report
Dear authors, thank you for the thorough revision. The manuscript is much more legible and presents clearer message to the reader.
I would suggest final minor corrections in the text as proof-read examination and enclosure of results before initiating the discussion of Figure 5.
Line 79 - 80: Figure 2 shows the metallographic pictures of different temperature samples under the optical microscope. It shows the metallographic pictures of samples 1, 2, and 3 from left to right.
Please correct to following:
“Figure 2 shows the metallographic pictures of different temperature samples (1, 2, and 3 from left to right) under the optical microscope.”
The explanation of XRD is much relevant now and illustrates occurrences of complex phase transformations.
In the inset or magnification of XRD plot in Figure 4 (b) from 40 – 50O, please put the label or peak identifiers on the minor peaks.
Please provide one liner explanation in relation to the phase diagram how α-Fe can form, so that the reader follows the overall analysis.
The answer you wrote to me in Response 15, I would suggest to incorporate here:
After line 126 – 127:
With the increase of temperature, the crystallite size of the sample becomes smaller. “Since the radius of Nd atom is larger than that of Fe and As atom, the vacancy number of Fe atoms is formed through thermal vibration. Therefore, the vacancy number increases with an increase in temperature. On the contrary, Nd tends to accumulate at the grain boundaries easily, preventing grains from growing and so grains are smaller.”
Line 127 – 128: Accordingly, high fraction of the boundary regions existed between Phase. ???
Please consider reader may not be very considerate to text. Kindly write here the phases (it cannot be a single phase for boundaries to coexist) and under which thermal conditions the grain boundaries will have different phases.
Line 136: First which adsorbs and then.
Correct it as following: “First As adsorbs and then”
Line 141: With the increase of temperature, the grain size smaller, the grain boundary increases, and there are,
Please correct it as following:
“With the increase in temperature, as the grain size becomes smaller, the grain boundary area increases, and there are”
Line 142 – 144: The combination of the vacancy and the Nd atom is the solute-vacancy complex, and the diffusion rate of the grain boundary is increased to cause the Nd to be concentrated in the grain boundary region.
Please modify it to:
“The combination of the vacancy and the Nd atom leads to the formation of the solute-vacancy complex, such that thermally enhanced diffusion rate of the boundaries causes Nd atoms to segregate in the grain boundary region.”
Please write about it as you mentioned in response 12:
“The darker phase in top part of Sample 1 in Figure 5 is in the upper part at the interface corresponds to phase formed due to the combination of arsenic and iron.”
Since the other details were out of the scope of this study, we can neglect the phase changes in the outer part of the sample.
FEG-SEM images and EDS results are much better now. I would have been happier if my viewpoints were elucidated from the first revision request:
Moreover, if we look at Figure 4, the grey phase is distributed very finely in the inside zone of sample 1, i.e. lower temperature. This phase tends to grow and encircle the darker greyish phase as in case of sample 2 and 3. Which phase is this? There are two different contrasts of fine grey phases here as well. Do you know which are they? If the brighter contrast is more evenly spread, it may be considered that Nd is diffusing e.g. sample 2, sparsely in sample 1 as the bright aggregates are larger.
If you look at the interface, there is a dark grey phase at the top part of sample 1. It should contain more Fe. This phase also disappears at the interface of sample 2 and 3. Do we expect some residual Fe left over at his temperature range in sample 1, such that vacancies are drawn as As sublimes (partially) and Nd atoms oscillate to these sites forming brighter contrast phases. The areal fraction of grey and dark grey phase is significantly less in image 2, what might be happening here as bright phase is spread across?
Darker or blackish phase also appear in sample 1 of Figure 5. Seldomly this phase appears in sample 3, but higher contrast means it is composed more of Fe.
But I can consider if the analysis has not been made previously, a detailed investigation may be out of scope of this revision.
What I would like is that one liner explanation is derived for Figure 5 (Line 132) to explain the results of all these three samples. You are only discussing here, without presenting the results. Talk about the results and then discuss them.
How the line scan indicates the variation in composition from Sample 1 to 3? How Nd, Fe and As derived binary phases vary in terms of scan length or phase contrast. This is most important part of your manuscript. These results should be mentioned before initiating the discussion.
Good luck.
Author Response
Dear Professor,
We are grateful to the editor and the referees for their constructive comments and suggestions on the revision of the manuscript again. We have made all the necessary changes as suggested by the referees. All the revisions in the manuscript and the revised information have been highlighted in red color.
Now, we are submitting the revised version of our manuscript.
Best regards,
Jinzhu Zhang
